# Prognostic Implication of Ventricular Volumetry in Early Brain Computed Tomography after Cardiac Arrest

**DOI:** 10.3390/diagnostics14161701

**Published:** 2024-08-06

**Authors:** Ae Kyung Gong, Sang Hoon Oh, Jinhee Jang, Kyu Nam Park, Han Joon Kim, Ji Young Lee, Chun Song Youn, Jee Yong Lim, Hyo Joon Kim, Hyo Jin Bang

**Affiliations:** 1Department of Emergency Medicine, Seoul St. Mary’s Hospital, College of Medicine, The Catholic University of Korea, Seoul 06591, Republic of Korea; kyung6069@naver.com (A.K.G.);; 2Department of Radiology, Seoul St. Mary’s Hospital, College of Medicine, The Catholic University of Korea, Seoul 06591, Republic of Korea

**Keywords:** heart arrest, brain ventricular volumetry, prognostication, neurological outcome

## Abstract

Brain swelling after cardiac arrest may affect brain ventricular volume. This study aimed to investigate the prognostic implications of ventricular volume on early thin-slice brain computed tomography (CT) after cardiac arrest. We measured the gray-to-white matter ratio (GWR) and the characteristics and volumes of the lateral, third, and fourth ventricles. The primary outcome was a poor 6-month neurological outcome. Of the 166 patients, 115 had a poor outcome. The fourth ventricle was significantly smaller in the poor outcome group (0.58 cm^3^ [95% CI, 0.43–0.80]) than in the good outcome group (0.74 cm^3^ [95% CI, 0.68–0.99], *p* < 0.001). Ventricular characteristics and other ventricular volumes did not differ between outcome groups. The area under the curve for the fourth ventricular volume was 0.68, comparable to 0.69 for GWR. Lower GWR (<1.09) and lower fourth ventricular volume (<0.41 cm^3^) predicted poor outcomes with 100% specificity and sensitivities of 8.7% (95% CI, 4.2–15.4) and 20.9% (95% CI, 13.9–29.4), respectively. Combining these measures improved the sensitivity to 25.2% (95% CI, 17.6–34.2). After adjusting for covariates, the fourth ventricular volume was independently associated with neurologic outcome. A marked decrease in fourth ventricular volume, with concomitant hypoattenuation on CT scans, more accurately predicted outcomes.

## 1. Introduction

Among patients who are successfully resuscitated from cardiac arrest, more than half never regain consciousness as a result of hypoxic–ischemic brain injury [1]. However, early identification of the severity of brain injury and prediction of subsequent neurological outcomes are challenging. Recent international guidelines for prognostication recommend a multimodal approach, including brain imaging [2,3]. Brain computed tomography (CT) can discriminate the cause of cardiac arrest and visualize the severity of brain injury [4,5]. Therefore, most intensivists consider brain CT to be one of the most useful tests [6].

Brain ischemia induces cerebral edema, resulting in the loss of distinction between gray and white matter on brain CT images [7,8]. The extent of brain edema can be quantified as the gray-to-white matter ratio (GWR), which has been extensively analyzed in arrest patients [9,10,11,12,13,14]. Low GWR seems to predict a poor outcome in comatose patients, particularly when combined with other prognostic methods.

Another ischemic change noted with brain CT is brain swelling. As brain injury progresses over time, brain swelling can lead to intracranial hypertension, altering the volumes of the intracranial components according to Monro–Kellie’s doctrine [15,16]. Therefore, brain injury after cardiac arrest can affect ventricular size and morphology. Although several investigators have analyzed ventricular characteristics or cerebrospinal fluid (CSF) volume on brain CT after cardiac arrest, the results are conflicting, and the prognostic importance of CT signs of brain swelling remains questionable [17,18,19]. Each ventricular volume (lateral, third, and fourth ventricles) can be differently affected, depending on the extent of brain injury [18,19]. Although standard CT scans of cross-sections are typically obtained using the reference line, sometimes failure in the process of adjusting the angle of the individual’s head in the CT gantry results in inconsistent axial brain images. Therefore, the simple characteristics of a single axial image may not accurately represent the real ventricle size, depending on the angulation of the CT scan.

In the present study, we hypothesized that a decrease in ventricular volume with or without concomitant hypoattenuation on CT scans would indicate severe brain injury. To evaluate their prognostic implications, we quantified brain edema and swelling using the GWR and ventricular volumetry from three-dimensional (3D) reconstructed images of the ventricles on early thin-slice brain CT scans after the return of spontaneous circulation (ROSC). Our primary aim was to quantify the ventricular volume and determine to what extent these changes corresponded to 6-month neurological outcomes. The secondary aim was to assess whether measures of ventricular volumetry can complement the GWR in predicting neurological outcomes.

## 2. Materials and Methods

### 2.1. Study Design and Patients

This registry-based observational study was conducted at one tertiary hospital. All the data were extracted from the Advanced Resuscitation and Outcome Prediction (AROP) registry between 2012 and 2022 [20]. The AROP registry was prospectively collected from consecutive OHCA patients. Our institutional ethics committee approved the study in 2019 (KC19OESI0191). The requirement for consent was waived for patients included until that point, after which written informed consent was obtained from each patient’s next of kin. The inclusion criteria were patients who were over 18 years of age, resuscitated from out-of-hospital cardiac arrest (OHCA), treated with targeted temperature management (TTM), and underwent brain CT scans after ROSC. We excluded patients without thin-slice brain CT images; with brain lesions that disturbed the analysis, such as artifacts, previous neurological lesions, and pseudo-subarachnoid hemorrhage (SAH); or whose brain CT data were missing.

### 2.2. Brain CT Acquisitions and Postresuscitation Care

During the study period, all patients routinely underwent nonenhanced brain CT scans, including thin-slice images, immediately after ROSC. However, patients who were transferred after brain CT scanning at the referring hospital and patients with hemodynamic instability did not undergo brain CT scans. In several patients, other CT scan protocols were used at the discretion of the attending physicians. Brain CT scans were performed using a 64-channel multidetector CT scanner (Somatom Sensation 64; Siemens Medical Solutions, Erlangen, Germany) with the following protocols: 120 kVp, 380 mAs, field of view = 250 × 250 mm, matrix 512 × 512, and slice thickness 0.6–0.75 mm. The clinical standard axial images were reconstructed with a slice thickness of 4 mm, a standard kernel for soft tissue, and a sharp kernel for the bone structures. In addition, thin-slice (0.6–0.75 mm) axial images were reconstructed with a standard kernel. After brain CT scanning, all patients were immediately treated with TTM at 33 °C or 36 °C for 24 h and postresuscitation care in accordance with local and international guidelines [2,3,21].

### 2.3. Brain CT Interpretation

The following CT data were collected from the AROP registry: the GWR in the basal ganglia and ventricular characteristics, including the distance of the anterior horn (the maximum width of the frontal horns of the lateral ventricles), Evan’s index (the ratio of the distance of the anterior horn and the maximal internal diameter of the skull), the maximal lateral diameter of the third ventricle and the anteroposterior diameter of the fourth ventricle [18,19].

In all axial slices, two investigators (S.H.O. and J.J.) who were blinded to patient outcomes independently segmented the three ventricles using the open-source and multiplatform 3D medical image analysis software ITK-SNAP [22]. Each ventricular volume was measured automatically, and average volumes were recorded. We also calculated three ventricular volume ratios: lateral ventricle/third ventricle, lateral ventricle/fourth ventricle, and third ventricle/fourth ventricle. The measurements of the GWR and ventricular volumes are summarized in Figure 1.

### 2.4. Outcome Measurement

The neurological outcome at 6 months after OHCA was evaluated in a face-to-face visit or a telephone interview with the patients or their relatives. The primary endpoint was a poor neurological outcome, defined as a cerebral performance category (CPC) score of 3–5. A CPC score of 1–2 indicated a good outcome.

### 2.5. Statistical Analysis

The categorical variables are displayed as a number (percentage), and the continuous variables are expressed as mean ± standard deviation or median and interquartile range (IQR). Comparisons of categorical variables between groups were made using the chi-squared test or Fisher’s exact test. After testing for a normal distribution, continuous variables were compared using Student’s *t*-test or the Mann–Whitney U test. We calculated the intraclass correlation coefficient (ICC) of ventricular volumes between two investigators. To assess the performances of the predictors, the receiver operating characteristic (ROC) curve, the cutoff values, and the sensitivities and specificities of the parameters, which were calculated using an exact binomial 95% confidence interval (CI), were evaluated. Pairwise area under the ROC curve (AUC) comparisons were made between predictors using the nonparametric approach developed by DeLong et al. [23]. To evaluate the association between ventricular volume and neurological outcome, we categorized patients into 4 quartiles of ventricular volume (Q1–Q4), and logistic regression analysis was used to estimate odds ratios (ORs) and 95% CIs.

All analyses were performed using SPSS 24.0 software (IBM, Armonk, NY, USA) and R version 4.3.2 (R Foundation for Statistical Computing, Vienna, Austria). A value of *p* < 0.05 was considered significant for all analyses.

## 3. Results

### 3.1. Characteristics of the Study Participants

A total of 326 adult OHCA patients were treated with TTM during the study period. Of these, 256 patients underwent brain CT, and 90 brain CT scans were excluded, leaving 166 patients for inclusion in the study (Appendix A). After 6 months, 51 (30.7%) patients had a good neurological outcome, and 115 (69.3%) patients had a poor neurological outcome. Coronary artery disease, shockable rhythm, and witnessed and cardiac etiology arrest were more common in the good outcome group than in the poor outcome group (all *p* < 0.05) (Table 1). The poor outcome group was older and had a longer arrest time (both *p* < 0.05).

### 3.2. Comparison of Brain CT between the Outcome Groups

The median ROSC-to-CT intervals in the good and poor outcome groups were 26.0 (IQR, 15.0–41.0) min and 14.0 (IQR, 10.0–29.0) min, respectively. The mean GWR was significantly lower in the poor outcome group (1.18 ± 0.07 vs. 1.23 ± 0.08, *p* < 0.001) (Table 1). None of the ventricular characteristics on axial imaging were significantly different between outcome groups (Table 1).

According to the analyses of ventricle volumetry, the ICCs between the investigators were good to excellent (Appendix A). The median fourth ventricular volume in the good outcome group was significantly greater than that in the poor outcome group (0.74 cm^3^ [0.68–0.99] vs. 0.58 cm^3^ [0.43–0.80], *p* < 0.001) (Table 1). Conversely, patients with good outcomes appeared to have smaller lateral ventricles than patients with poor outcomes, although the difference was not statistically significant (20.69 cm^3^ [13.32–28.20] vs. 25.93 cm^3^ [14.42–36.75], *p* = 0.214). The third ventricular volume was similar between the good and poor outcome groups (1.02 cm^3^ [0.75–1.59] vs. 1.26 cm^3^ [0.65–1.73], *p* = 0.919). The lateral-to-fourth ventricular volume ratio was significantly different between outcome groups (32.7 [IQR, 16.1–38.7] vs. 44.5 [IQR, 23.1–61.6], *p* = 0.006).

### 3.3. Prognostic Performance of the GWR Alone and Ventricular Volume Alone

The AUC for the prediction of a poor outcome was calculated to evaluate the predictive performance of each predictor on brain CT (Figure 2). The AUC of the fourth ventricular volume (0.68 [95% CI, 0.61–0.75]) was comparable to that of the GWR (0.69 [95% CI, 0.61–0.76]). On the other hand, the prognostic performances of ventricular characteristics and lateral and third ventricular volume were poor (Figure 2A,B), although differences between the GWR and fourth ventricular volume were not statistically significant (all *p* > 0.05).

When the cutoff values with 100% specificity for predicting poor outcome were analyzed, GWR < 1.09 predicted a poor outcome with a sensitivity of 8.7% (95% CI, 4.2–15.4) (Table 2). Fourth ventricular volume < 0.41 cm^3^ predicted a poor outcome with a sensitivity of 20.9% (95% CI, 13.9–29.4) while maintaining 100% specificity. Lower lateral (<7.30 cm^3^) or third ventricular volume (<0.49 cm^3^) also indicated poor outcomes, with sensitivities of 7.8% and 16.5%, respectively.

We analyzed the performance of combinations of the GWR and each ventricular volume as poor outcome predictors (GWR < 1.09, lateral ventricle < 7.30 cm^3^, third ventricle < 0.49 cm^3^ or fourth ventricle volume < 0.41 cm^3^) (Figure 3). A substantial number of patients with poor outcomes and GWR ≥ 1.09 had small lateral, third, and fourth ventricles (*n* = 7, *n* = 17, and *n* = 19, respectively). Finally, when these small ventricle volumes were added to the GWR, the sensitivities improved to 14.8% (95% CI, 8.9–22.6), 23.5% (95% CI, 16.1–32.3), and 25.2% (95% CI, 17.6–34.2), respectively, while maintaining 100% specificity.

### 3.4. Multivariable Logistic Regression Models for Predicting Poor Outcomes

Because of the small sample and in accordance with the rule of 10, five variables were included in the multivariable logistic regression models (Table 3) [24]. When adjusting for variables associated with ventricular volume (such as age, sex, and CT acquisition time), smaller fourth ventricular volume was an independent predictor of poor outcomes; the ORs of Q1, Q2 and Q3 (quartiles of fourth ventricle volume) were 4.86 (95% CI, 1.42–16.70), 2.85 (95% CI, 0.92–8.82) and 0.70 (95% CI, 0.27–1.87), respectively. However, in the multivariable model, including various resuscitation variables, neither the GWR nor fourth ventricular volume was independently associated with neurological outcome.

### 3.5. Relationships between Lateral and Fourth Ventricular Volumes and Outcomes

Appendix A shows the alluvial plots of the subjects in the good and poor outcome groups divided into small (Q1–Q2) and large (Q3–Q4) lateral and fourth ventricular volumes. Among the patients who were categorized as having a large fourth ventricle, those with a large lateral ventricle were significantly more likely to have a poor outcome than those with a small lateral ventricle (28/42 vs. 16/41, *p* = 0.012), and they were older than the others (63.7 ± 13.5 vs. 48.4 ± 13.6, *p* < 0.001). In the analysis of patients with small fourth ventricles, patients with large lateral ventricles were older than those with small lateral ventricles (66.8 ± 13.1 vs. 46.9 ±13.5, *p* < 0.001), but their poor outcome ratios were similarly poor (34/41 vs. 37/42, *p* = 0.503).

## 4. Discussion

In this study, we analyzed the GWR, ventricular characteristics, and volumes in early brain CT scans after ROSC. Ventricular characteristics did not differ between the good and poor 6-month neurological outcome groups. The AUC of the fourth ventricular volume for predicting poor neurological outcome was comparable to that of the GWR, and the combination of two measurements that predict poor outcome with 100% specificity increased the sensitivity of predicting poor outcome with 100% specificity.

Hypoxic–ischemic brain injury affects neuronal ion pumps and induces cytotoxic and vasogenic edema [25], which manifests as reduced GWR and eventually leads to brain swelling. These imaging findings develop over time in patients with severe brain injuries, and thus, brain injuries may not be fully reflected in early brain CT scans [26,27,28]. According to our analysis of brain CT scans obtained immediately after ROSC, the sensitivity of the GWR for predicting poor outcomes was only 8.7%. These findings are consistent with recent studies that acquired early CT scans, wherein the GWR predicted poor outcomes with relatively low sensitivity (3.5–20.3%) [9,10,29].

Measuring brain swelling is another objective tool for quantifying cerebral edema [30]. This change may manifest later than the reduced GWR, which is indicative of cellular edema, as it represents a secondary change in increased tissue volume due to brain edema [19,31]. In our findings, the group with poor outcomes had a significantly smaller fourth ventricle, while their lateral and third ventricles tended to be larger. These disparities in ventricular volume by neurological outcomes can be explained by the anatomy of the ventricular system. The cerebral aqueduct, the narrowest portion connecting the third and fourth ventricles, originates near the basal ganglia and is particularly vulnerable to brain injury in the early stages after cardiac arrest [32]. Consequently, it can easily become obstructed, leading to impaired drainage of CSF [18,33], and then the size of the fourth ventricle may decrease while the lateral and third ventricles remain unchanged or can even expand. CSF empties from the fourth ventricle into the cisterns of the skull base and then into the lumbar CSF space and the subarachnoid space at the sagittal sinus. Therefore, similar to the fourth ventricle, the distance of the posterior ambient cistern can serve as a predictor in early brain CT scans following cardiac arrest [34]. However, the differences in ventricular characteristics in our study did not reach statistical significance, which is consistent with previous studies that reported inconsistent findings [18,19]. We hypothesize that ventricular characteristics measured on a single axial image, especially the anteroposterior diameter, may be shortened or lengthened depending on the axis of the CT scan. This study is the first to measure each ventricular volume from consecutive images, and we believe that volumetric measurements can be promising prognostic tools with high measurement reliability. Previous studies have shown that the CSF volume proportion did not differ based on neurological outcomes in patients scanned shortly after ROSC [17,35]. However, considering the decreased proportion of CSF observed on delayed CT scans (3–4 days after cardiac arrest) in patients with poor outcomes [35], the total CSF volume, unlike the fourth ventricle volume, may have limited utility as an outcome predictor in early brain CT scans before brain swelling becomes evident.

The absence of a cistern has been identified as a poor outcome predictor in children [36]. Although our study excluded patients with pseudo-SAH with prominent ominous signs, a considerable proportion of patients with poor outcomes showed marked reductions in ventricular volume. Twenty-four patients had a fourth ventricular volume < 0.41 cm^3^ and exhibited a poor outcome, whereas thresholds of the lateral and third ventricles displayed lower sensitivities. Our findings suggest that instead of a dichotomous classification, physicians should consider very small ventricular volumes as indicators of poor outcomes. Ventricular volumes are known to correlate with increasing age due to atrophic changes in the brain parenchyma [37,38]. Given the importance of age as a covariate in outcome prediction [39], distinguishing pathologic decreases in ventricle size from normal age-related changes is crucial. Longitudinal CT data, including both pre- and arrest phases, are optimal for study purposes. However, the age-related increase in the fourth ventricle is less pronounced than that in other ventricles [40,41]. After adjusting for age, sex, GWR, and CT scan timing, a small fourth ventricle was found to be an independent predictor of poor outcomes.

Current prognostic guidelines recommend assessing the presence of generalized brain edema, but the GWR is the only quantitative measure of brain CT that predicts poor neurological outcome [2,3]. Distortion of normal anatomy in patients with severe brain injury and cerebral edema may raise concerns regarding GWR measurements depending on visual assessment [19]. However, distinguishing CSF from brain parenchyma is easy due to apparent differences between Hounsfield units. Measuring ventricular volumetry demonstrated a prognostic performance similar to that of the GWR. Interestingly, substantial discrepancies were observed between the GWR and ventricular volume in each patient. Accordingly, a combination of markedly reduced GWR and fourth ventricular volume improved the sensitivity (25.2%), with 100% specificity for predicting poor neurological outcome.

The present study has several limitations. First, it was a single-center study with a relatively small cohort of patients. The wide CIs observed in our results reflect this limited sample size. Second, as mentioned above, the time from ROSC to brain CT scan may be too short to fully exhibit brain structural changes, potentially restricting the generalizability of our findings to cohorts with different CT acquisition timings. Third, although fourth ventricular volume emerged as a predictor of neurological outcome even after adjusting for covariates such as age, ventricular size can vary significantly between individuals. Hence, we propose a reduced threshold value for ventricular volume to achieve a false-positive rate of 0% in predicting a poor outcome. Given these limitations, our results must be interpreted with caution and require further confirmation in larger prospective studies. Additional research on changes in ventricular volume as brain injury develops over time through serial brain CT scans is warranted.

## 5. Conclusions

In this analysis of early brain CT scans after cardiac arrest, fourth ventricular volume was independently associated with poor outcomes after adjusting for covariates, including age, and its prognostic performance was comparable to that of the GWR. Combining marked decreases in fourth ventricular volume with the GWR improved the sensitivity for identifying poor outcomes compared to either single measurement while maintaining 100% specificity.

## Figures and Tables

**Figure 1 diagnostics-14-01701-f001:**
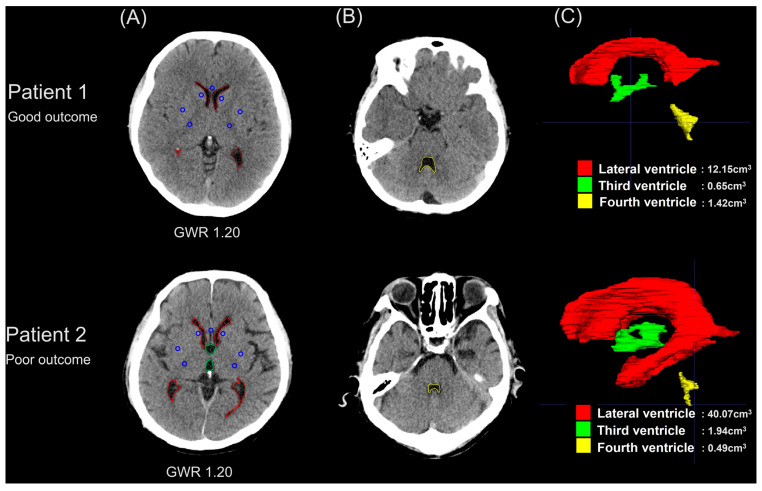
Brain computed tomography images and 3D reconstructed ventricle images from two of the patients. The upper panel shows images from 21-year-old female patients with good neurological outcomes, and the lower panel shows images from 72-year-old male patients with poor neurological outcomes. (**A**,**B**) Blue circular regions of interest (10 mm^2^) were bilaterally located on the corpus callosum (CC), caudate nucleus (CN), putamen (PU), and posterior limb of the internal capsule (PIC) at the basal ganglia level, and the gray-to-white matter ratio (GWR) was calculated from each Hounsfield unit according to the following formula: GWR = (CN + PU)/(CC + PIC). Red, green, and yellow lines indicate outlines of the lateral, third, and fourth ventricles on axial images, respectively. (**C**) Red, green, and yellow images indicate 3D reconstructed images of the lateral, third, and fourth ventricles, respectively.

**Figure 2 diagnostics-14-01701-f002:**
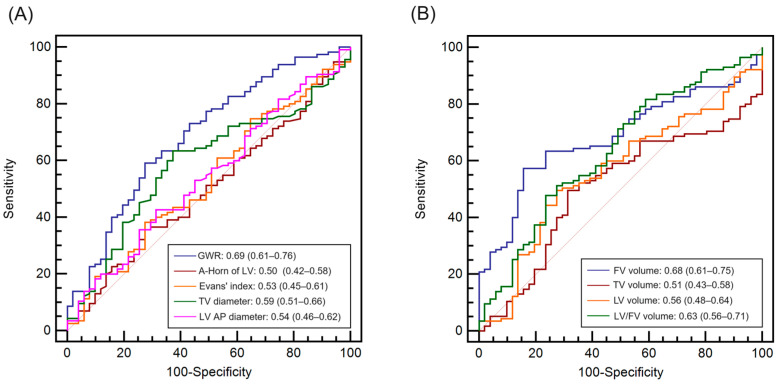
Receiver operating characteristic curve for the prediction of a poor 6-month neurological outcome. (**A**) The AUCs and 95% CIs for the GWR and ventricular characteristics (distance of the anterior horns of the lateral ventricle, Evans’ index, lateral distance of the TV, and the AP diameter of the FV). (**B**) The AUCs and 95% CIs for the volume of the lateral ventricle, third ventricle, fourth ventricle, and lateral ventricle/fourth ventricle volume ratio. AUC, area under the curve; CI, confidence interval; GWR, gray-to-white matter ratio; A-Horn, anterior horns; LV, lateral ventricle; TV, third ventricle; FV, fourth ventricle; AP, anterior-posterior.

**Figure 3 diagnostics-14-01701-f003:**
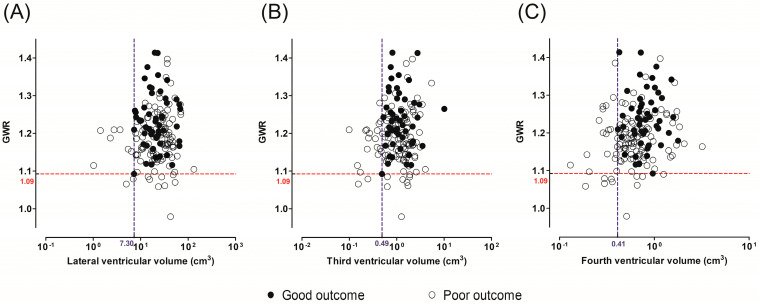
Scatter plots showing the distribution of the GWR (gray-to-white matter ratio) and ventricular volumes in the good and poor neurological outcome groups. The X-axes are on a log scale. Red and blue dotted lines indicate cutoff values for 100% specificity for poor outcomes. (**A**) GWR and lateral ventricular volume. (**B**) GWR and third ventricular volume. (**C**) GWR and fourth ventricular volume.

**Table 1 diagnostics-14-01701-t001:** Baseline characteristics of the included patients.

	Good Outcome (*n* = 51)	Poor Outcome (*n* = 115)	*p*-Value
Male	38 (74.5)	75 (65.2)	0.236
Age, years, mean ± SD	51.7 ± 14.4	58.6 ± 16.3	0.010
Coronary artery disease	14 (27.5)	11 (9.6)	0.003
Hypertension	21 (41.2)	49 (42.6)	0.863
Diabetes mellitus	7 (13.7)	30 (26.1)	0.077
Chronic renal failure	4 (7.8)	9 (7.8)	1.000
Witnessed	41 (80.4)	71 (62.3)	0.021
Bystander CPR	38 (74.5)	73 (64.0)	0.185
Shockable rhythm	40 (78.4)	31 (27.0)	<0.001
Cardiac etiology	48 (94.1)	56 (48.7)	<0.001
Time from arrest to ROSC, min, median (IQR)	16.6 (9.0–20.0)	36.7 (24.0–42.0)	<0.001
Target temperature, 33 °C	45 (88.2)	102 (88.7)	0.932
ROSC-to-CT interval, min, median (IQR)	26.0 (15.0–41.0)	14.0 (10.0–29.0)	<0.001
GWR, mean ± SD	1.23 ± 0.08	1.18 ± 0.07	<0.001
Ventricular characteristics			
Distance between the anterior horns, mm	37.1 (34.7–40.7)	36.7 (34.3–40.9)	0.993
Evans’ index	0.277 (0.255–0.301)	0.278 (0.260–0.308)	0.525
Lateral diameter of TV, mm, median (IQR)	6.0 (4.9–7.6)	7.0 (5.0–8.9)	0.072
Anteroposterior diameter of FV, mm, median (IQR)	6.2 (4.9–7.8)	6.5 (5.2–7.9)	0.410
Ventricular volumetry			
LV, cm^3^, median (IQR)	20.69 (13.32–28.20)	25.93 (14.42–36.75)	0.214
TV, cm^3^, median (IQR)	1.02 (0.75–1.59)	1.26 (0.65–1.73)	0.919
FV, cm^3^, median (IQR)	0.74 (0.68–0.99)	0.58 (0.43–0.80)	<0.001
LV/TV, median (IQR)	19.5 (14.9–23.2)	22.6 (15.8–28.3)	0.066
LV/FV, median (IQR)	32.7 (16.1–38.7)	44.5 (23.1–61.6)	0.006
TV/FV, median (IQR)	1.9 (0.9–2.0)	2.1 (1.0–2.8)	0.066

Data are presented as *n* (%) for the categorical variables unless otherwise indicated. SD, standard deviation; CPR, cardiopulmonary resuscitation; ROSC, return of spontaneous circulation; IQR, interquartile range; CT, computed tomography; GWR, gray-to-white matter ratio; Evans’ index; the ratio of the maximum width of the frontal horns of the lateral ventricles and the maximal internal diameter of the skull; LV, lateral ventricle; TV, third ventricle; FV, fourth ventricle.

**Table 2 diagnostics-14-01701-t002:** Prognostic accuracies of the predictors from early brain computed tomography for identifying poor 6-month neurological outcomes.

	Cutoff	TP	FP	TN	FN	Sensitivity (95% CI)	Specificity (95% CI)
GWR	<1.20 ^a^	68	14	37	47	59.1 (49.6–68.2)	72.6 (58.3–84.1)
	<1.09 ^b^	10	0	51	105	8.7 (4.2–15.4)	100.0 (93.0–100.0)
Distance between the anterior horns, mm	>39.4 ^a^	41	14	37	74	35.7 (26.9–45.1)	72.6 (58.3–84.1)
	>47.5 ^b^	3	0	51	112	2.6 (0.5–7.4)	100.0 (93.0–100.0)
Evans’ index	>0.297 ^a^	44	14	37	71	38.3 (29.4–47.8)	72.6 (58.3–84.1)
	>0.357 ^b^	3	0	51	112	2.6 (0.5–7.4)	100.0 (93.0–100.0)
Lateral diameter of TV, mm	>6.31 ^a^	73	19	32	42	63.5 (54.0–72.3)	62.8 (48.1–75.9)
	>11.70 ^b^	5	0	51	110	4.4 (1.4–9.9)	100.0 (93.0–100.0)
Anteroposterior diameter of FV	>6.97 ^a^	49	16	35	66	42.6 (33.4–52.2)	68.6 (54.1–80.9)
	>11.08 ^b^	4	0	51	111	3.5 (1.0–8.7)	100.0 (93.0–100.0)
Volume of LV, cm^3^	>26.17 ^a^	57	14	37	58	49.6 (40.1–59.0)	72.6 (58.3–84.1)
	<7.30 ^b^	9	0	51	106	7.8 (3.6–14.3)	100.0 (93.0–100.0)
Volume of TV, cm^3^	>1.34 ^a^	57	16	35	58	49.6 (40.1–59.0)	68.6 (54.1–80.9)
	<0.49 ^b^	19	0	51	96	16.5 (10.3–24.6)	100.0 (93.0–100.0)
Volume of FV, cm^3^	<0.61 ^a^	66	8	43	49	57.4 (47.8–66.6)	84.3 (71.4–93.0)
	<0.41 ^b^	24	0	51	91	20.9 (13.9–29.4)	100.0 (93.0–100.0)
LV volume/FV volume	>38.7 ^a^	55	12	39	60	47.8 (38.4–57.3)	76.5 (62.5–87.2)
	>119.5 ^b^	4	0	51	111	3.5 (1.0–8.7)	100.0 (93.0–100.0)

Data are presented as *n* for categorical variables unless otherwise indicated. ^a^ The cutoff value selected by the Youden index. ^b^ The cutoff value had 100% specificity for determining a poor neurological outcome. TP, true positive; FP, false positive; FN, false negative; TN, true negative; CI, confidence interval; GWR-BG, gray-to-white matter ratio at the basal ganglia level; Evans’ index, the ratio of the maximum width of the frontal horns of the lateral ventricles to the maximal internal diameter of the skull; LV, lateral ventricle; TV, third ventricle; FV, fourth ventricle.

**Table 3 diagnostics-14-01701-t003:** Multivariable logistic models for independent predictors associated with poor neurological outcome at 6 months.

	Univariable		Multivariable I ^a^		Multivariable II ^b^	
	OR (95% CI)	*p*-Value	AOR (95% CI)	*p*-Value	AOR (95% CI)	*p*-Value
Male	0.64 (0.31–1.34)	0.238	0.77 (0.32–1.87)	0.571	N.A.	
Age, per year	1.03 (1.01–1.05)	0.012	1.04 (1.01–1.06)	0.009	N.A.	
Nonshockable rhythm	9.85 (4.50–21.59)	<0.001	N.A.		2.99 (0.79–11.26)	0.107
Noncardiac etiology	16.86 (4.97–57.24)	<0.001	N.A.		15.26 (2.59–89.81)	0.003
Arrest time, per min	1.10 (1.07–1.14)	<0.001	N.A.		1.13 (1.07–1.18)	<0.001
ROSC to CT, per min	1.00 (0.99–1.00)	0.196	1.00 (0.99–1.00)	0.666	N.A.	
GWR						
Quartile 1	5.10 (1.84–14.12)	0.002	4.18 (1.37–12.73)	0.012	1.13 (0.23–5.41)	0.883
Quartile 2	4.46 (1.67–11.94)	0.003	4.06 (1.34–12.36)	0.014	1.90 (0.43–8.43)	0.401
Quartile 3	1.89 (0.78–4.55)	0.156	2.33 (0.85–6.42)	0.102	1.66 (0.41–6.70)	0.474
Quartile 4	Ref.		Ref.		Ref.	
FV volume						
Quartile 1	5.64 (1.84–17.28)	0.002	4.86 (1.42–16.70)	0.012	1.82 (0.42–7.90)	0.425
Quartile 2	3.91 (1.41–10.84)	0.009	2.85 (0.92–8.82)	0.069	4.55 (1.00–20.69)	0.050
Quartile 3	0.78 (0.33–1.86)	0.578	0.70 (0.27–1.87)	0.481	0.28 (0.06–1.22)	0.089
Quartile 4	Ref.		Ref.		Ref.	

^a^ Ventricular volume-related variable model: ORs are adjusted for sex, age, CT acquisition time, GWR, and FV volume. ^b^ Resuscitation variable-related model: ORs are adjusted for nonshockable rhythm, noncardiac cause, arrest time, GWR, and FV volume. OR, odds ratio; AOR, adjusted odds ratio; CI, confidence interval; CPR, cardiopulmonary resuscitation; ROSC, return of spontaneous circulation; CT, computed tomography.

## Data Availability

The data presented in this study are available on request from the corresponding author. The data are not publicly available due to legal restrictions.

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
