# Peer review of "Prognostic Implication of Ventricular Volumetry in Early Brain Computed Tomography after Cardiac Arrest"

_diagnostics, 2024, doi:10.3390/diagnostics14161701_

Round 1

Reviewer 1 Report

Comments and Suggestions for Authors

The authors of the manuscript present results from an original registry-based observational study, conducted at a tertiary hospital, including 166 patients who suffered a cardiac arrest. They aimed to assess the prognostic value of brain ventricles volume measurement, performed by early thin-slice brain computed tomography (CT) in these patients. The primary outcome for this study was poor neurological outcome at Month 6th.

The study is topical and important not only from scientific, but also from clinical point of view. Cardiac arrests account for 15-20% of all natural deaths in adults in Europe, the USA and other countries and up to 50% of all cardiovascular deaths. To reduce this burden, more knowledge is needed about its incidence in various world regions, risk factors (populations at increased risk), prognostic indicators for patients, who have already sufferd a cardiac arrest, etc. For this reason, any clinical study that adds more information about this topic is valuable. According to the presented results in current manuscript  4th cerebral ventricular volume has been independently associated with neurologic outcome: a marked decrease of the volume of this brain ventricle with concomitant hypoattenuation on CT scans predicted accurately the primary outcomes defined for this study.

The manuscript is generally well-structured, the text is plain and readable. 

I have the following recommendations to the authors, which I believe will improve additionally the quality of their work:

1. The sections of the Abstract have to be clearly separated for convenience of the readers: Background....; Aims (objectives) of the study; Materials and methods.....; Results....; Conclusions....;

2. Clear inclusion/exclusion criteria should be defined and presented in "Materials and methods"

Reviewer 2 Report

Comments and Suggestions for Authors

The paper by Gong et al. aims to investigate the prognostic implications of ventricular volume on early brain CT after cardiac arrest. The main contributions include identifying that a smaller fourth ventricular volume is significantly associated with poor 6-month neurological outcomes. Additionally, the study demonstrates that combining this measure with the gray-to-white matter ratio (GWR) improves sensitivity for predicting poor outcomes while maintaining high specificity. The paper is well-written, with a sufficiently detailed introduction, adequately described methods, clearly presented results, conclusions that are supported by the findings, and limitations that are well described.

I have only two minor comments:

1. Why have you only used the GWR central and not peripheral (e.g., frontal, parietal, temporal, occipital)? Consider using it as well to investigate if this is an additional marker for the prediction of poor outcomes. It would enhance the quality of your paper.

2. Page 3, line 114 (legend of figure 1): Was your circular ROI really 10 cm² or rather 1 cm²?
